# Postpartum Weight Change in Relation to Pre-Pregnancy Weight and Gestational Weight Gain in Women in Low-Income Setting: Data from the KITE Cohort in the Northern Part of Ethiopia

**DOI:** 10.3390/nu14010131

**Published:** 2021-12-28

**Authors:** Kebede Haile Misgina, Henk Groen, Afework Mulugeta Bezabih, Hendrika Marike Boezen, Eline M. van der Beek

**Affiliations:** 1Department of Public Health, College of Health Sciences, University of Aksum, Axum P.O. Box 1010, Ethiopia; 2Department of Epidemiology, University Medical Center Groningen, University of Groningen, 9700 RB Groningen, The Netherlands; h.groen01@umcg.nl (H.G.); h.m.boezen@umcg.nl (H.M.B.); 3School of Public Health, College of Health Sciences, University of Mekelle, Mekelle P.O. Box 231, Ethiopia; afework.mulugeta@gmail.com; 4Department of Paediatrics, University Medical Center Groningen, University of Groningen, 9700 RB Groningen, The Netherlands; e.m.van.der.beek@umcg.nl

**Keywords:** postpartum weight change, postpartum maternal nutrition, postpartum weight retention, pre-pregnancy weight, gestational weight gain

## Abstract

(1) Background: Postpartum weight may increase compared to pre-pregnancy due to weight retention or decrease due to weight loss. Both changes could pose deleterious effects on maternal health and subsequent pregnancy outcomes. Therefore, this study aimed to assess postpartum weight change and its associated factors. (2) Methods: A total of 585 women from the **KI**lte-Awlaelo **T**igray **E**thiopia (KITE) cohort were included in the analysis. (3) Results: The mean pre-pregnancy body mass index and weight gain during pregnancy were 19.7 kg/m^2^ and 10.8 kg, respectively. At 18 to 24 months postpartum, the weight change ranged from −3.2 to 5.5 kg (mean = 0.42 kg [SD = 1.5]). In addition, 17.8% of women shifted to normal weight and 5.1% to underweight compared to the pre-pregnancy period. A unit increase in weight during pregnancy was associated with higher weight change (β = 0.56 kg, 95% CI [0.52, 0.60]) and increased probability to achieve normal weight (AOR = 1.65, 95% CI [1.37, 2.00]). Food insecurity (AOR = 5.26, 95% CI [1.68, 16.50]), however, was associated with a shift to underweight postpartum. Interestingly, high symptoms of distress (AOR = 0.13, 95% CI [0.03, 0.48]) also negatively impacted a change in weight category. (4) Conclusions: In low-income settings such as northern Ethiopia, higher weight gain and better mental health during pregnancy may help women achieve a better nutritional status after pregnancy and before a possible subsequent pregnancy.

## 1. Introduction

Women gain weight during pregnancy, driven mainly by normal physiological changes and tissue remodeling. The optimal weight increase depends on pre-pregnancy body mass index (BMI). To this end, specific weight gain ranges according to pre-pregnancy BMI categories are recommended by the Institute of Medicine (IOM) guidelines [1]. After pregnancy, women are supposed to return to their pre-pregnancy weight, usually by the sixth postpartum week. However, many studies show that 20 to 50% of women retain a substantial part of the weight gained during pregnancy at six months to two years postpartum [2,3,4,5,6,7,8]. High postpartum weight retention may contribute to obesity and adverse health outcomes, including chronic non-communicable diseases [9,10,11]. Postpartum weight retention or weight gain may also complicate subsequent pregnancies [11,12,13,14]. On the other hand, in low-income countries where undernutrition is widespread, retaining some weight after pregnancy may be translated into better nutritional status (a change from underweight to normal weight category). Therefore, it may have an advantageous effect on maternal and child health outcomes [15].

Postpartum weight change, defined as the difference between postpartum weight and weight before pregnancy, may be predicted by different factors. These include parity [16], pre-pregnancy body mass index [2,4], gestational weight gain (GWG) [2,3,6,17,18,19,20,21,22,23,24], physical activity [4], breastfeeding [4,24,25,26,27], psychosocial characteristics [28,29], and dietary factors [30]. However, most studies focused solely on assessing the association between pre-pregnancy body mass index, gestational weight gain, and postpartum weight, overlooking potentially important effects of socioeconomic characteristics, reproductive and obstetric conditions, psychosocial factors, and food and dietary habits [2,3,18,19,20,21,22,23]. In addition, previous studies were mainly conducted in developed countries, where pre-pregnancy overweight and excessive gestational weight gain are major public health problems. For instance, approximately 40%, 45%, and 30% of women in the USA, UK, and Europe, respectively, enter pregnancy being overweight [1,22,31,32,33]. At the same time, nearly 50% of women in the USA, UK, and Europe gain more weight during pregnancy than recommended [3,22,32,33].

Unlike the situation in developed countries, the foremost concern in low-income countries is pre-pregnancy undernutrition and gaining inadequate weight during pregnancy [34]. Therefore, studies on postpartum weight change in low-income countries may have to be aimed at different outcomes than in developed countries. However, no study has assessed postpartum weight change in low-income countries where it may manifest as excess weight retention or excess weight loss. A change to both sides of the spectrum could pose significant public health concerns because of their deleterious effects on maternal health and subsequent birth outcomes. In addition, most previous studies used self-reported weight(s) to compute postpartum weight change [18], instead of actual weight measurements before and after pregnancy. As a result, previous findings may be biased, and a knowledge gap concerning actual postpartum weight change and its determinants remains for low-income countries. Therefore, our study aimed to assess measured instead of self-reported postpartum weight change in relation to pre-pregnancy weight and gestational weight gain. We investigated a wide range of factors with respect to their association with postpartum weight change in a population of pregnant women in the northern part of Ethiopia.

## 2. Methods

### 2.1. Study Design, Setting, and Population

The present study was part of the **KI**lte-Awlaelo **T**igray **E**thiopia (KITE) cohort study designed to assess maternal nutrition, adverse birth outcomes, and child growth, as reported previously [35]. The KITE cohort, a population-based prospective study, was conducted between February 2018 and September 2020 in the Kilte-Awlaelo Health and Demographic Surveillance Site (KA-HDSS) in the eastern zone of the Tigray region in northern Ethiopia. The KA-HDSS comprises thirteen kebeles (the smallest administrative unit). The total population of KA-HDSS is about 110,000. Most of the population lives in rural conditions, where agriculture is the primary source of income.

The study population and measurements were described in detail previously [35]. The study population included pregnant women whose delivery date lay before February 2019. To be eligible for the study, pregnant women needed to be married, aged 18 or older, had pre-pregnancy weight measurements, and completed ≤20 weeks of gestation at inclusion. The sample size was calculated to address the objectives of the KITE cohort study, which was to demonstrate a difference of 8% in low birth weight, depending on mid-upper arm circumference (MUAC) (above vs. below 23.0 cm). The additional parameters used were alpha = 0.05 and beta = 0.20. Including an estimated 10% drop-out rate, the total sample size was calculated at 1100. 

The weight of non-pregnant women (*n* = 17,500) living in the study area was measured using a SECA scale to the nearest 100 g between August and October 2017. Subsequently, eligible women were identified by applying different methods including a community-based survey by Health Extension Workers through the “Women’s Development Army”, a network of health information workers reaching individual households through their health posts. Further, the records of the nearby antenatal clinics and the KA-HDSS database were used. All eligible pregnant women identified between February and September 2018 were included consecutively. In total, 991 women were recruited for the KITE cohort study. Out of the 991 women, 585 were followed until 18 to 24 months postpartum. After the index pregnancy, women who became pregnant again within the follow-up period were excluded from the present analysis.

### 2.2. Data Collection and Measurements

Qualified Health Extension Workers collected the data via an interviewer-administered questionnaire and anthropometric measurements. The data collection was supplemented by extracting data available in the KA-DHSS database and prenatal records. As the details of the measures are accessible elsewhere [35,36], only brief descriptions are provided below.

**Anthropometric data:** Weight (to the nearest 100 g) using a SECA scale, height (to the nearest 0.1 cm) using a height-measuring board, and MUAC (to the nearest 0.1 cm) using a non-stretchable tape were measured at inclusion. Also, weight and MUAC were measured during pregnancy at 32 to 36 weeks of gestation, and 18 to 24 months postpartum. All measurements were obtained twice and averaged. Similarly, pre-pregnancy BMI, BMI at inclusion, and postpartum BMI were computed from height measured at inclusion and weight measured before pregnancy, at inclusion, and 18 to 24 months postpartum, respectively, as BMI = weight (kg)/[height (m)]^2^. Then, based on the BMI (in kg/m^2^), women were classified as underweight (BMI < 18.5 kg/m^2^), normal (BMI = 18.5 to 24.9 kg/m^2^), and overweight (BMI ≥ 25.0 kg/m^2^). Gestational weight gain calculated as the difference between weights at 32 to 36 weeks of gestation and before pregnancy was classified based on the IOM guidelines [1]. Moreover, postpartum weight change was calculated by subtracting pre-pregnancy weight from the weight measured 18 to 24 months postpartum.

**Sociodemographic data:** Age, residence, education, occupation, household size, and economic status were extracted from the KA-DHSS database. KA-DHSS updates the database every six months. However, the regular biannual update of the KA-DHSS database does not include economic status. Hence, adjustments for a change in the socioeconomic proxy indicator variables, since the last updates were made at inclusion. Wealth index quintiles designating the lowest to the highest economic statuses were generated from the proxy indicators. These indicators included housing characteristics, access to improved water and sanitation facilities, and ownership of household assets, land, and livestock [37]. 

Access to improved drinking water sources refers to access to piped water on-premises, public taps or standpipes, tube wells or boreholes, protected dug wells, protected springs, and/or rainwater collection. Similarly, access to an improved sanitation facility was defined as access to an unshared toilet facility, pit latrine with a slab, ventilated improved pit latrine, or flush toilet [38].

Health extension package implementation was assessed at inclusion by checking if the women’s respective households were certified as models. A model household was defined as a household trained on a health extension package and that implemented the package afterwards [39,40,41]. Additionally, self-reported physical activity was collected at inclusion using the short form of the International Physical Activity Questionnaire (IPAQ) and summarized according to the scoring protocol [42,43]. Moreover, self-reported history of pre-pregnancy illnesses was collected at inclusion.

**Reproductive and obstetric conditions:** Gravidity and parity were extracted from the KA-DHSS database. In addition, self-reported intimate partner violence was obtained at inclusion using the four-item HITS (Hurt, Insult, Threaten, and Scream) questions, each rated from 1 to 5. A summed score >10 was used as suggestive of violence [44]. Additionally, women were asked nine questions addressing five domains relating to women’s empowerment at inclusion: earning and control over income, decision-making on household purchases, mobility and healthcare autonomy, attitude towards intimate partner violence, and ownership of assets [45,46,47]. Coding each response as 0 or 1 and summing all the responses, a women’s empowerment score ranging from 0 to 9 was obtained. By assigning the domains an equal weight of 1 point each, to be shared by the indicators within the domains, women who scored ≥80% were considered empowered [48]. Additionally, women were queried at inclusion if they wanted to get pregnant at the time they became pregnant, later, or not at all. Accordingly, an index pregnancy that was wanted later or not wanted at all was considered unplanned. 

Moreover, self-reported stressful life events that occurred over the past year, illness during pregnancy, pregnancy complications, prenatal care, and complications at birth were obtained at 32 to 36 weeks or at or immediately after birth, as appropriate [45,49,50]. For women who began prenatal care at ≤16 weeks of gestation, prenatal care was defined as adequate plus (five or more visits), adequate (four visits), or intermediate (two to three visits). Prenatal care was considered inadequate if started at >16 weeks and/or comprising fewer than two visits [51]. Apart from the self-reported history of illness during pregnancy, data on HIV status, urine analysis, stool examination, venereal diseases, hepatitis B, haemoglobin, and other diseases were retrieved from prenatal records when available. Based on the measurement at prenatal care booking, iron deficiency anaemia was defined as haemoglobin <11 g/dL [52]. Self-reported duration of exclusive breastfeeding was also obtained at 18 to 24 months postnatal.

**Food and dietary data:** At inclusion, 24 h of dietary diversity data were collected by asking women about a list of food groups, with ‘yes’ or ‘no’ response options. Consumed foods were grouped into ten categories: grains, white roots and tubers; pulses; nuts and seeds; dairy; meat, fish, and poultry; egg; dark green leafy vegetables; other vitamin-A rich fruit and vegetables; other fruit; and other vegetables. A score of consuming at least five groups was defined as adequate dietary diversity [53]. Additionally, data on fasting were collected by asking women if they participate in the weekly fast and adhere to the long fast times. Finally, women were categorized as fasting if they fasted both weekly and for the long fasting times.

Food insecurity was assessed using the Household Food Insecurity Access Scale, consisting of nine ‘yes’ or ‘no’ occurrence questions. Each positive response was followed by a frequency-of-occurrence question, asking how often—(1) rarely, (2) sometimes, or (3) often—the reported food insecurity-associated condition happened in the previous month. Then, the sum of the frequency-of-occurrence questions yielded a food insecurity score ranging from 0 to 27. If the responses to all occurrence questions were ‘no’ or if the affirmative response was only to “did you worry that your household would not have enough food” in a rare frequency of occurrence, households were classified as food secure [54].

**Psychosocial characteristics data:** The ten-item Edinburgh Postnatal Depression Scale (EPDS) [55] and the seven-item anxiety subscale of the Hospital Anxiety and Depression Scale (HADS-A) [56] were used to measure depression and anxiety. Each item in both scales was rated from 0 to 3. For stress, the four-item Perceived Stress Scale (PSS-4), scored from 0 to 4, was used [57]. All were measured at 32 to 36 weeks of gestation. The HADS-A score ≥8, EPDS score of ≥13, and PSS-4 ≥ 8 were defined as high depression, anxiety, and stress symptoms, respectively. Total perinatal distress was calculated as the sum of anxiety, stress, and depression scores. To indicate the level of distress, the presence of high symptoms in one, two, or all of the three domains, i.e., anxiety, depression, or stress, was considered.

The five-item Turner Support Scale measured partner support; each scored from 0 to 3, with scores of <10 indicating low support [58]. Support from significant other social sources was also rated using the Oslo-3 Social Support Scale, and a score ≤8 was considered low [59]. Both measures of support at inclusion were summed up as a total support score. Additionally, low total social support was defined as low support from both partner and other social sources. 

### 2.3. Statistical Analysis

Data were entered to Epi-Data (Version 3.1, EpiData Association, Denmark. Http://www.epidata.dk accessed on 30 June 2018), verified by re-entering 20% of the completed questionnaires selected randomly, and analyzed with Stata (Version 14, Stata Corporation, and College Station, Texas, USA). Proportions and means (SD) or medians (IQRs) were used to describe the characteristics of the participants. In addition, baseline characteristics were compared between women who successfully completed the study and those who did not using a Chi-squared test, an independent Student’s t-test, or a Mann–Whitney U-test (as appropriate) to evaluate selective loss to follow-up. Additionally, the unadjusted association of the independent variables with postpartum weight change as a continuous score was estimated using univariable linear regression.

To assess factors independently influencing postpartum weight change, a hierarchical multivariable linear regression modeling was applied. In the first model, all statistically significant independent variables (*p* < 0.05, tested two-sided) from the univariable analysis, except pre-pregnancy body mass index and gestational weight gain, were included. Further, an interaction between food insecurity and perinatal distress was included as per the likelihood ratio test in the first model. Next, pre-pregnancy body mass index was added. In the third and final model, gestational weight gain was included. A hierarchical modeling approach was selected to evaluate the relevance of the independent variables. Normality of residuals was assessed through the normal probability plot and quantile–quantile plot. Homogeneity of variance was checked using the Breusch–Pagan test. In addition, multicollinearity was tested using variance inflation factor (vif).

Furthermore, logistic regression was used to identify factors associated with a change in BMI category between the prenatal and the postnatal period. In this case, the dependent variables were shifting from pre-pregnancy underweight to normal postnatal weight and shifting from pre-pregnancy normal weight to postnatal underweight. All statistically significant independent variables (*p* < 0.05, tested two-sided) from the univariable logistic regression analysis were included in the respective multivariable logistic regression analysis.

We performed inverse probability-weighted analyses to examine the possibility of selective loss to follow-up. First, we fitted a probit model with follow-up status (successfully completed the study vs. lost to follow-up) as a dependent variable and baseline characteristics as independent variables. The baseline characteristics included residence, age, parity, education, occupation, being a model household, history of illness, food insecurity, diet diversity, physical activity, pre-pregnancy BMI, and GWG. Then, we calculated the predicted probability of successfully completing the study from the probit model. Finally, the inverse of the predicted probability was used to weight the observations in a re-analysis of factors associated with postpartum weight change [60].

## 3. Results

From the 991 women included at an average of 14.8 weeks (SD = 1.9) of gestation, 585 were followed until 18 to 24 months postpartum. The women who were lost to follow-up did not differ from women who completed the study and were included in the present analysis for most of the baseline characteristics. However, women who were lost to follow-up were more likely to be urban dwellers, housewives, and have had a history of illness during pregnancy. On average, the women included in the present analysis were 29.6 years (SD = 6.4) old, and the majority (69.2%) lived in rural conditions. Education-wise, about a third (32.7%) had at most a primary school education. By occupation, 353 (59.1%) were farmers and 180 (30.2%) were housewives. Moreover, more than half of the women (51.3%) reported low physical activity. Moreover, their average parity, including the index birth, was 3.8 (SD = 2.3). Regarding the women’s health condition, 111 (18.6%) had a history of illness during the index pregnancy, and 44.4% had high symptoms of distress at least in one of the three domains (Table 1).

Before pregnancy, the mean body mass index of the 585 women included in the present study was 19.7 kg/m^2^ (SD = 2.0). On average, the women gained 10.8 kg (SD = 2.3) over the course of pregnancy. At 18 to 24 months postpartum, their pre-pregnancy weight changed from −3.2 to 5.5 kg (Figure 1), with the mean change being 0.42 kg (SD = 1.5). As a result, the average body mass index increased to 19.9 kg/m^2^ (SD = 2.2). In terms of a change in weight category, 17.8% of women who were underweight before pregnancy achieved normal weight, whereas 5.1% shifted to underweight compared to the pre-pregnancy period. In total, 194 (33.2%) women were underweight at 18 to 24 months postpartum (Appendix A). The postpartum weight change in kg in relation to pre-pregnancy body mass index and gestational weight gain is shown in Figure 2.

In the univariable analysis, women empowerment, dietary diversity, social support, adequacy of prenatal care, pre-pregnancy BMI, and gestational weight gain were positively associated with postpartum weight change. Additionally, physical activity, food insecurity, and perinatal distress were negatively related with postpartum weight change. In the first multivariable model, women empowerment (β [95% CI] 0.11 [0.01, 0.20]), adequate prenatal care (0.49 [0.22, 0.77]), and high physical activity (−0.52 [−1.04, −0.001]) retained their association. When model 1 was additionally adjusted for pre-pregnancy BMI in model 2, adequate prenatal care (0.48 [0.21, 0.74]) and pre-pregnancy BMI (0.14 [0.07, 0.22]) were associated with postpartum weight change. After adjusting model 1 for both pre-pregnancy weight and weight gain during pregnancy in model 3, an average of 0.56 kg (0.52, 0.60) was retained at 18 to 24 months postpartum for each kilogram of increase in gestational weight gain (Table 2).

Table 3 shows factors associated with a change in weight category at 18 to 24 months postpartum in relation to pre-pregnancy weight. Food insecurity (AOR [95% CI] 5.26 [1.68, 16.50]) was associated with shifting from normal pre-pregnancy weight to underweight 18 to 24 months postpartum. A 1 kg decrease in gestational weight gain (AOR 2.38 [1.64, 3.45]) was associated with higher odds of shifting from normal pre-pregnancy weight to underweight at 18 to 24 months postpartum. Similarly, being distressed in two domains (AOR 0.13 [0.03, 0.48]) was associated with lower odds of shifting to the normal weight category. Conversely, a 1 kg increase in gestational weight gain was associated with shifting from pre-pregnancy underweight to the normal weight category (AOR 1.65 [1.37, 2.00]).

Because of the loss to follow-up, we re-analyzed the data weighted with the inverse probability of successfully completing the study to examine the effect of selection bias on effect estimates and the observed associations. Of the baseline characteristics considered in the analysis, only a history of illness during pregnancy (AOR = 0.60, 95% CI [0.44, 0.83]) was associated with the probability of successfully completing the study. Furthermore, we did not see any difference between the weighted and unweighted analyses with regard to the effect estimates and observed associations described above (data not shown).

## 4. Discussion

In the present study, we have assessed postpartum weight change in relation to pre-pregnancy weight and gestational weight gain and its associated factors among women in the northern part of Ethiopia. In our study population, the average weight change at 18 to 24 months postpartum was 0.42 kg (SD = 1.5), much lower than reported in the existing literature [61,62]. The disagreement with the literature is likely explained by the difference in the socioeconomic and nutritional characteristics of the women. In our study, a sizable number of the women were food insecure and had inadequate diet diversity. Moreover, 36.4% of the women were underweight before pregnancy and 62.2% did not achieve adequate gestational weight gain [35], unlike in the previously reported studies. At 18 to 24 months postpartum, about one third (33.2%) of the women were still underweight. This implies that the postpartum period is an underutilized window of opportunity to improve maternal nutritional status, and possibly child health, in our study area. 

In terms of a change in weight category, 17.8% of women who were underweight before pregnancy had achieved normal weight at 18 to 24 months postpartum. Gaining more weight during pregnancy within the range recommended by the IOM guideline was associated with a higher postpartum weight change [62,63] and shifting from lower to normal weight category. These results show that adequate gestational weight gain is essential to optimize maternal health in low-income countries where pre-pregnancy under-nutrition is prevalent. In addition, the women who shifted from low to normal weight may have improved maternal reserve after pregnancy. This improved maternal reserve may help women achieve optimal lactation without depleting their nutrient stores and be better prepared for a possible next pregnancy. Thus, adequate gestational weight gain may contribute to enhancing subsequent pregnancy outcomes and child health. As shown in our previous work, interventions that advance women’s empowerment, dietary quality, pre-pregnancy nutritional status, and prenatal care utilization may improve gestational weight gain [36].

With specific reference to shifting to a lower weight category, 5.1% of women moved from normal pre-pregnancy weight to underweight at 18 to 24 months postpartum. In low-income countries, postpartum weight loss compounded with inadequate weight gain during pregnancy, and even more in a situation of pre-pregnancy undernutrition, might negatively impact women’s health. In addition, postpartum weight loss leading to being underweight might also affect outcomes of subsequent pregnancies. Therefore, postpartum weight status deserves the attention of the health care professionals in low-income countries. During pregnancy, gaining less weight was corroborated with postpartum weight loss and shifting from a higher (normal) to a lower weight category. This finding implies that evidence-driven interventions geared to achieve adequate gestational weight gain are needed in low-income countries like Ethiopia. Our findings may also suggest the need for post-pregnancy nutritional interventions as part of the maternal care continuum to optimize maternal health and subsequent pregnancy outcomes.

Furthermore, food insecurity was associated with higher odds of becoming underweight in the postpartum period among normal-weight women before pregnancy. The observed association may be partly explained by inadequate dietary intake due to lack of access to food. The relationship between food insecurity and being underweight postpartum may also show the need for nutritional interventions focusing on food-insecure women/households. Similarly, distress was associated with a lower likelihood of shifting from underweight to normal weight. Clearly, distress was associated with remaining underweight from before pregnancy to about two years postpartum. The effect of distress on appetite, nutrient/food intake, and dietary choices may partly explain the observed association. Although the link between food insecurity and distress can be bidirectional, the impact of food insecurity on distress may also aggravate the situation. Therefore, the relation between distress and remaining underweight from before pregnancy to about two years postpartum may highlight the need to tailor perinatal interventions and better screen and manage maternal mental health.

No previous study assessed factors associated with a shift in weight after pregnancy compared to pre-pregnancy, i.e., from normal to underweight or underweight to normal weight. Previous studies reported a positive association between food insecurity and distress and net postpartum weight change [64,65,66], exactly opposite to our finding. The most likely explanation for the disagreement in the results is the differences in characteristics of the study population. For example, food insecurity and distress may affect predominantly food choices (i.e., high energy low nutrient density foods) in developed countries and instead lead to more postpartum weight retention [67]. In some cases, distress may also be followed by emotional eating and corroborate with more weight retention. On the contrary, food insecurity in low income countries is all about limited access to adequate food that impairs both the quality and quantity of diet/foods/nutrients in our study population. Furthermore, socioeconomic adversity such as food insecurity is high in low-income countries like Ethiopia and may cause psychological distress and worsen its influence on maternal diet and nutrient status.

To our knowledge, this is the first study on this topic in sub-Saharan Africa. The results highlight the need to understand and map the problem to develop more targeted interventions. However, the follow-up time in the present study was short. Future studies with longer follow-up of the women after pregnancy may be needed to gain better insight into long-term risks and benefits. These long-term risks and benefits may be evaluated in terms of maternal health and child growth as well as outcomes of subsequent pregnancies. Of note, our subgroup analyses assessing factors associated with a change in weight category may not be adequately powered. Furthermore, data on potential confounding factors such as physical activity and dietary characteristics were only collected at inclusion, early during pregnancy, and may not adequately reflect the situation during the postnatal period. Therefore, future studies with a higher sample size and including data on potential confounders after pregnancy are recommended.

## 5. Conclusions

Our findings suggest that higher weight gain and better mental health during pregnancy may help women achieve a better nutritional status after pregnancy and before a possible subsequent pregnancy. Hence, in low-income settings such as northern Ethiopia, where maternal undernutrition is common, adequate support during pregnancy may improve maternal and child health outcomes.

## Figures and Tables

**Figure 1 nutrients-14-00131-f001:**
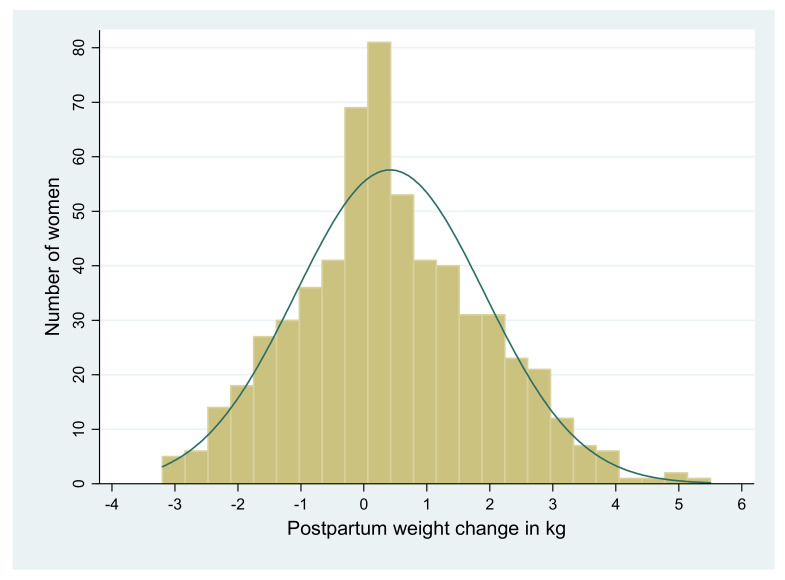
Distribution of postpartum weight change in kg among 585 women from the Tigray region, northern Ethiopia, 2018.

**Figure 2 nutrients-14-00131-f002:**
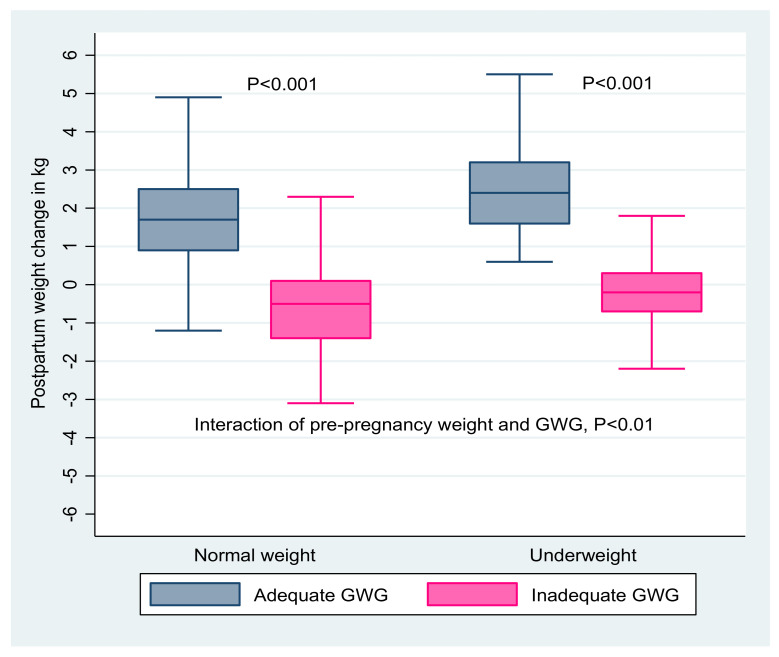
Postpartum weight change in kg separated by pre-pregnancy BMI (left panel: normal weight: BMI 18 to 25; right panel: underweight: BMI < 18.5) and gestational weight gain (GWG, adequate in blue, inadequate in pink) in Tigray region, northern Ethiopia, 2018. *p*-values indicate comparisons of adequate versus inadequate GWG among normal weight and underweight women.

**Table 1 nutrients-14-00131-t001:** Socio-demographic factors, reproductive and obstetric conditions, food and dietary habits, and psychosocial characteristics.

Characteristics	Baseline *n* = 991	Lost to Follow-Up *n* = 406	Completed the Study, *n* = 585	*p*-Value ^1^
Age at inclusion, mean (SD)	29.3 (6.5)	28.9 (6.5)	29.6 (6.5)	0.111
<25 years	273 (27.6)	120 (29.6)	153 (26.1)	0.478
25–34 years	490 (49.5)	197 (48.5)	293 (50.1)	
≥35 years	228 (23.0)	89 (21.9)	139 (23.8)	
Rural by residence, *n* (%)	647 (65.3)	244 (60.1)	403 (68.9)	**0.004**
Educational status, *n* (%)				0.186
No formal education	362 (36.5)	136 (33.5)	226 (38.6)	
Primary education	326 (32.9)	135 (33.3)	191 (32.7)	
Secondary education and above	303 (30.6)	135 (33.3)	168 (28.7)	
Occupational status, *n* (%) ^$^				**0.005**
Farmer	541 (54.6)	197 (48.5)	344 (58.8)	
Housewife	337 (34.0)	159 (39.2)	178 (30.4)	
Others *	113 (11.4)	50 (12.3)	63 (10.8)	
Economic status				0.845
Lowest	198 (20.0)	84 (20.7)	114 (19.5)	
Low	198 (20.0)	81 (20.0)	117 (20.0)	
Middle	200 (20.2)	83 (20.4)	117 (20.0)	
High	200 (20.2)	85 (20.9)	115 (19.7)	
Highest	195 (19.6)	73 (18.0)	122 (20.8)	
Household size, mean (SD)	5.5 (2.0)	5.3 (2.0)	5.6 (2.0)	**0.027**
Model household, *n* (%)	242 (24.4)	106 (26.1)	136 (23.3)	0.303
History of pre-pregnancy illness, *n* (%)	142 (14.3)	56 (13.8)	86 (14.7)	0.688
Level of physical activity, *n* (%)				0.058
Low	527 (53.2)	226 (55.7)	301 (51.4)	
Moderate	425 (42.9)	159 (39.1)	266 (45.5)	
High	39 (3.9)	21 (5.2)	18 (3.1)	
Total support score, mean (SD)	21.3 (3.8)	21.3 (3.9)	21.3 (3.7)	0.944
Low social support, *n* (%)	75 (7.6)	38 (9.4)	37 (6.3)	0.413
Improved source of drinking water, *n* (%)	888 (89.6)	367 (90.4)	521 (89.1)	0.499
Improved sanitation facility, *n* (%)	135 (13.6)	51 (12.6)	84 (14.4)	0.417
Number of stressful life events, mean (SD)	0.6 (0.9)	0.7 (1.0)	0.6 (0.9)	0.161
Total perinatal distress score (*n* = 938), mean (SD)	19.0 (9.2)	19.0 (9.2)	19.1 (9.2)	0.890
Level of perinatal distress (*n* = 938), *n* (%)				
Not distressed at all	532 (56.7)	207 (58.6)	325 (55.6)	0.792
Distressed in one domain	201 (21.4)	71 (20.1)	130 (22.2)	
Distressed in two domains	118 (12.6)	42 (11.9)	76 (13.0)	
Distressed in three domains	87 (9.3)	33 (9.4)	54 (9.2)	
Total parity, mean (SD) ^$^	3.7 (2.3)	3.4 (2.2)	3.8 (2.3)	**0.013**
Primigravida	196 (19.8)	88 (21.7)	108 (18.5)	**0.038**
Two to three	341 (34.4)	154 (37.9)	187 (32.0)	
Four to five	239 (24.1)	89 (28.9)	150 (25.6)	
More than five	215 (21.7)	75 (18.5)	140 (23.9)	
Unplanned pregnancy, *n* (%)	405 (40.9)	161 (39.7)	244 (41.7)	0.518
Intimate partner violence score, mean (SD)	6.9 (3.0)	6.9 (3.1)	6.9 (3.0)	0.933
Experienced intimate partner violence, *n*(%)	161 (16.3)	75 (18.5)	86 (14.7)	0.113
Women empowerment score, mean (SD)	5.6 (1.5)	5.5 (1.6)	5.6 (1.4)	0.202
Low women empowerment, *n* (%)	114 (11.5)	47 (11.6)	67 (11.5)	0.952
Adequacy of prenatal care use, *n* (%)				0.829
Inadequate	418 (42.2)	170 (41.9)	248 (42.4)	
Intermediate	126 (12.7)	56 (13.8)	70 (12.0)	
Adequate	389 (39.3)	158 (38.9)	231 (39.5)	
Adequate plus	58 (5.9)	22 (5.4)	36 (6.1)	
History of illness during pregnancy, *n* (%)	225 (22.7)	116 (28.6)	109 (18.6)	**0.000**
Number of pregnancy complications, mean (SD)	1.2 (1.4)	1.3 (1.4)	1.2 (1.5)	0.270
Number of complications at birth, mean (SD)	0.4 (0.8)	0.4 (0.9)	0.4 (0.8)	0.259
Dietary diversity score, mean (SD)	4.6 (1.4)	4.7 (1.4)	4.5 (1.4)	0.051
Adequate dietary diversity score, *n* (%)	518 (52.3)	219 (53.9)	299 (51.1)	0.380
Food insecurity score, median (IQR)	0 (0–8)	0 (0–8)	0 (0–8)	0.604 ^a^
Food insecure, *n* (%)	392 (39.6)	155 (38.2)	237 (40.5)	0.459
Fasting, *n* (%)	694 (70.0)	290 (71.4)	404 (69.1)	0.423
Exclusive breastfeeding in months, mean (SD)	5.4 (1.2)	—	5.4 (1.2)	—
Exclusive breastfeeding (*n* = 585), *n* (%)	439 (73.5)	—	439 (73.5)	—

^1^ For comparisons between women who were lost to follow-up and those who successfully completed the study, * Students, unemployed, and so on, ^$^ The difference between women who were lost to follow-up and those who successfully completed the study was insignificant after stratification by residence. *p*-Values written in bold show significant difference.

**Table 2 nutrients-14-00131-t002:** Linear regression analysis of factors associated with postpartum weight change in kg.

Characteristics	Univariable Analysis	*p*-Value	Multivariable Model 1 ^a^	*p*-Value	Multivariable Model 2 ^b^	*p*-Value	Multivariable Model 3 ^c^	*p*-Value
	Unadj. β (95% CI)		Adj. β (95% CI)		Adj. β (95% CI)		Adj. β (95% CI)	
Women empowerment score	0.13 (0.04, 0.21)	0.003	0.11 (0.01, 0.20)	**0.033**	0.07 (−0.03, 0.17)	0.156	−0.01 (−0.07, 0.05)	0.848
Food insecurity score	−0.03 (−0.05, −0.01)	0.015	−0.03 (−0.10, 0.04)	0.399	−0.003 (−0.03, 0.04)	0.856	0.02 (−0.004, 0.04)	0.729
Dietary diversity score	0.11 (0.02, 0.21)	0.014	0.04 (−0.07, 0.15)	0.451	0.02 (−0.09, 0.13)	0.760	0.02 (−0.04, 0.08)	0.107
Total physical activity								
Low	Reference	-	Reference	-	Reference	-	Reference	-
Moderate	−0.26 (−0.51, −0.01)	0.043	−0.22 (−0.47, 0.04)	0.102	−0.13 (−0.39, 0.15)	0.340	0.01 (−0.15, 0.16)	0.949
High	−0.65 (−1.13, −0.17)	0.008	−0.52 (−1.04,−0.001)	**0.049**	−0.42 (−0.95, 0.10)	0.115	0.02 (−0.36, 0.37)	0.967
Total social support score	0.04 (0.003, 0.07)	0.030	0.01 (−0.03, 0.05)	0.661	0.02 (−0.02, 0.06)	0.350	−0.000 (−0.02, 0.02)	0.973
Total perinatal distress score	−0.01 (−0.03,−0.001)	0.042	−0.004 (−0.02, 0.02)	0.683	0.03 (−0.05, 0.10)	0.445	−0.002 (−0.05, 0.04)	0.922
Adequacy of prenatal care								
Inadequate	Reference	-	Reference	-	Reference	-	Reference	-
Intermediate	0.32 (−0.08, 0.71)	0.116	0.24 (−0.16, 0.63)	0.240	0.21 (−0.18, 0.61)	0.293	0.08 (−0.18, 0.33)	0.545
Adequate	0.54 (0.27, 0.81)	0.000	0.49 (0.22, 0.77)	**0.000**	0.47 (0.20, 0.74)	**0.001**	0.04 (−0.13, 0.21)	0.672
Adequate plus	0.12 (−0.40, 0.65)	0.645	−0.04 (−0.59, 0.51)	0.88	−0.14 (−0.67, 0.39)	0.613	−0.14 (−0.45, 0.18)	0.371
Pre-pregnancy BMI in kg/m^2^	0.17 (0.11, 0.23)	0.000	Not included	-	0.15 (0.07, 0.23)	**0.000**	−0.06 (0.52, 0.60)	0.078
Gestational weight gain in kg	0.53 (0.50, 0.57)	0.000	Not included	-	Not included	-	0.56 (0.52, 0.60)	**0.000**
** Adjusted R^2^**	**Not applicable**		**5.6%**		**8.2%**		**66.3%**	

^a^ adjusted for interaction between total perinatal distress and food insecurity scores, ^b^ Model 1 + pre-pregnancy BMI, and ^c^ Model 1 + pre-pregnancy BMI + gestational weight gain. Unadj.; unadjusted, Adj; adjusted and BMI; body mass index. *p*-Values written in bold show significant association.

**Table 3 nutrients-14-00131-t003:** Factors associated with a change in BMI category at 18 to 24 months postpartum.

Characteristics	Total, *n* = 372	Shifted to Underweight ^1^	COR (95% CI)	*p*-Value	AOR (95% CI)	*p*-Value
Parity including the index birth, *n* (%)						
Primigravida	73 (19.6)	3 (15.8)	0.37 (0.10, 1.44)	0.151	1.18 (0.21, 6.70)	0.854
Two to three	116 (31.2)	3 (15.8)	0.23 (0.06, 0.86)	0.030	0.24 (0.04, 1.53)	0.130
Four to five	98 (26.3)	4 (21.1)	0.36 (0.11, 1.21)	0.099	0.35 (0.08, 1.51)	0.161
More than five	85 (22.9)	9 (47.4)	Reference	-	Reference	-
Unplanned index pregnancy, *n* (%)	146 (39.3)	12 (8.2)	2.80 (1.07, 7.29)	0.035	3.06 (0.89, 10.51)	0.076
Intimate partner violence, *n* (%)	39 (10.5)	6 (15.4)	4.43 (1.58, 12.56)	0.005	3.19 (0.77, 13.23)	0.111
Food insecurity, *n* (%)	95 (25.5)	13 (13.7)	7.08 (2.61, 19.22)	0.000	5.26 (1.68, 16.50)	**0.004**
Inadequate dietary diversity, *n* (%)	136 (36.6)	12 (8.8)	3.16 (1.21, 8.25)	0.019	3.19 (0.90, 11.34)	0.074
GWG in kg, mean (SD)	11.4 (2.1)	8.6 (1.3)	2.17 (1.67, 2.78) *	0.000	2.38 (1.64, 3.45) *	**0.000**
**Characteristics**	**Total, *n* = 213**	**Shifted to Normal Weight ^2^**	**COR (95% CI)**	***p*-Value**	**AOR (95% CI)**	***p*-Value**
Rural in residence, *n* (%)	162 (76.1)	24 (14.8)	0.46 (0.22, 0.98)	0.043	0.55 (0.19, 1.56)	0.260
Educational status, *n* (%)						
No formal education	97 (45.5)	13 (13.4)	Reference	-	Reference	-
Primary education	67 (31.5)	19 (28.4)	2.56 (1.16, 5.64)	0.020	2.38 (0.86, 6.65)	0.097
Secondary and above	49 (23.0)	6 (12.2)	0.90 (0.32, 2.54)	0.845	0.52 (0.12, 2.33)	0.394
Perinatal distress, *n* (%)						
Not distressed at all	76 (35.7)	21 (27.6)	Reference	-	Reference	-
Distressed in one domain	58 (27.2)	10 (17.2)	0.55 (0.23, 1.28)	0.162	0.38 (0.14, 1.08)	0.068
Distressed in two domains	40 (18.8)	3 (7.5)	0.21 (0.06, 0.77)	0.018	0.13 (0.03, 0.48)	**0.002**
Distressed in three domains	39 (18.3)	4 (10.3)	0.30 (0.10, 0.95)	0.040	0.38 (0.10, 1.46)	0.158
GWG in kg, mean (SD)	9.7 (2.1)	11.4 (2.1)	1.63 (1.37, 1.94)	0.000	1.65 (1.37, 2.00)	**0.000**

COR; crude odds ratio, AOR; adjusted odds ratio, ^1^ shifted from normal weight to underweight compared to pre-pregnancy, ^2^ shifted from underweight to normal compared to pre-pregnancy, and * COR and AOR reversed. *p*-Values written in bold show significant association.

## Data Availability

Dataset will be available on reasonable request from the authors.

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
