# Peer review of "Postpartum Weight Change in Relation to Pre-Pregnancy Weight and Gestational Weight Gain in Women in Low-Income Setting: Data from the KITE Cohort in the Northern Part of Ethiopia"

_nutrients, 2021, doi:10.3390/nu14010131_

Round 1

Reviewer 1 Report

The topic of the study is interesting and could be of interest in looking into maternal health in low-income countries. It is suggested that the study is divided into 2 parts. The study collected many data at baseline, but the data set was only partly utilised and this doesn’t stop the authors to present the results and discussed them independently (part 1).

Due to the fact that some study participants dropped out, the second half of the paper can compare the cohort who has completed the study, thus making the paper less confusing.

The methods section can be made clearer, the terms used in the results section must be introduced here. When was the inclusion stage occurred?

Please correct all grammatical errors and write clearly. The captions can be more descriptive, so they can be easily understood when reading them by themselves.

Abstract.

The introduction can be improved when the specific aims of the study are included, and they are related to the study outcome.

Introduction

L 49 insert citations

L 57 clarify which factors were overlooked in literature

The aim of the study must be clearly stated by mentioning which health factors were monitored or which outcomes were observed besides the gestational weight gain.

Methods

L 81-87 mention the ethics approval for the clinical study

L 89 what does ‘briefly’ mean here?

L 91 what were the exclusion criteria? Which pregnancy was reported here? First, second or which one?

How was the BMI calculated when only body weight measurements were obtained?

L 89-99 this paragraph is lacking details and must be rewritten.

L 107 what does inclusion refer to here? Use a term commonly used.

L 118-132 the writing is not clear at all, correct the grammar please

L 152 insert the citation for ‘anemia’ and clarify which ‘iron deficiency’ was monitored

L 190 insert the source of epi data v 3.1.

L 217-225 were confounding factors considered when analysing the data?

Results

Figure 1 place the y-axis title

Figure 2 caption must be rewritten to make it clearer

Table 1. Pre-pregnancy BMI was classified into underweight and normal weight. Explain and justify the reason not to tabulate the data based on this sub-cohort.

Some parameters are better express in range rather than mean values.

Table 2

Please clarify whether the baseline characteristics by separating the cohorts into participants completed the study and those dropped out to avoid any skewed data.

The footnote must be rewritten to improve understanding, what do participating an non-participating women correspond to?

Many qualitative data were collected in the study but only listed out in the tables. The characteristics of the study participants must be included as con-founding factors, this will lead to more meaningful data interpretation.

Discussion

L 294 how many? N=?

L 296 which studies? Insert citations and were the study cohorts comparable?

L 297-297 the statement must include other data, e.g diet diversity, anthropometric and other characteristics before a suggestion was made.

L 300-324 as previous comment, data interpretation must take into consideration of other characteristics that could play a role in the weight change.

L 346 insert citation

L 355-357 please check for accuracy of the sentence

L 361 the baseline data can analyse the confounding factor and see whether they contribute to any difference in the data set.

Authors are advised to clearly describe what were the limitations of the study and suggest improvements in future study.

The conclusion must focus on comprehensively analysed data as per previous comments.

Author Response

Reviewer 1: Comments and suggestions for authors

Comment: The topic of the study is interesting and could be of interest in looking into maternal health in low-income countries. It is suggested that the study is divided into 2 parts. The study collected many data at baseline, but the data set was only partly utilised and this doesn’t stop the authors to present the results and discussed them independently (part 1). Due to the fact that some study participants dropped out, the second half of the paper can compare the cohort who has completed the study, thus making the paper less confusing.

Response: Thank you for your feedback. Indeed, we are presenting data from the total cohort (n=991) as well as for the subgroup of women who had complete follow-up with respect to postpartum weight change (n=585). To assess the possibility of bias due to loss to follow-up, we have performed inverse probability weighting and compared the results of weighted and unweighted analyses. This analysis has been added to the manuscript (L 318-324).

Comment: The methods section can be made clearer; the terms used in the results section must be introduced here. When was the inclusion stage occurred?

Response: As some of the details of the methods used for data collection are provided in a published manuscript (Misgina, K.H, et al. BMJ Open 2021, 11, e043484.), some terms were not introduced again. To avoid confusion, we have introduced the relevant terms (variables) in the revised version now (L 141-146). The actual inclusion period is also mentioned in the revised text with some additional details about the inclusion process (L 102-111).

Comment: Please correct all grammatical errors and write clearly. The captions can be more descriptive, so they can be easily understood when reading them by themselves.

Response: Thank you for your suggestion. We have carefully revised the captions. Also, we have reread the paper and corrected grammatical errors.

Abstract

Comment: The introduction can be improved when the specific aims of the study are included, and they are related to the study outcome.

Response: Our outcome was postpartum weight change and we wanted to investigate which factors influence it. We reformulated our aim as follows: “Therefore, this study aimed to examine postpartum weight change and factors that influence it”.

Introduction

Comment: L 49 insert citations

Response: Thank you for your suggestion. We have cited the appropriate reference (L 52).

Comment: L 57 clarify which factors were overlooked in literature

Response: Thank you for your comment. Several socioeconomic characteristics, reproductive and obstetric conditions, psychosocial factors and food and dietary habits were overlooked. This is now clarified in the revised text (L 59-60).

Comment: The aim of the study must be clearly stated by mentioning which health factors were monitored or which outcomes were observed besides the gestational weight gain.

Response: The socioeconomic characteristics, reproductive and obstetric conditions, psychosocial factors and food and dietary habits that were overlooked in the previous studies are considered in the present study (L 78-82).

Methods

Comment: L 81-87 mention the ethics approval for the clinical study

Response: Thank you for your comment. According to the author’s guide, we think that it should be mentioned towards the end of the manuscript. Therefore, we included it at the end of the manuscript (L 418-421).

Comment: L 89 what does ‘briefly’ mean here?

Response: As the details of the KITE cohort can be found elsewhere (Misgina, K.H, et al. BMJ Open 2021, 11, e043484.), we have provided only a short summary in this manuscript. Therefore, ‘briefly’ was meant to indicate that all the details are not provided. Where relevant, we have provided more details (L 93).

Comment: L 91 what were the exclusion criteria? Which pregnancy was reported here? First, second or which one?

Response: At inclusion (recruitment), only married women, aged 18 or older, who had pre-pregnancy weight measurements and women who completed ≤20 weeks of gestation were included (L 95-97). Therefore, unmarried women, women younger than 18, whose weight was not measured, and those who completed >20 weeks of gestation were excluded. This is for the baseline data collection. Additionally, at 18 to 24 months after the index pregnancy, some women were found to be pregnant again. Therefore, these women were excluded from the present analysis. We clarified this in the revised version (L 110-111).

Comment: How was the BMI calculated when only body weight measurements were obtained?

Response: Thank you for your comment. Height was measured at inclusion or recruitment into the cohort and this height measurement was used to calculate BMI from weight measurements at several timepoints (before pregnancy, at inclusion, and at 18 to 24 months postpartum). The women were 18 years or older, so we do not expect a change in height between the measurements of weight. Weight measurement was added to the specification of anthropometric data (L 102-103, 118-127).

Comment: L 89-99 this paragraph is lacking details and must be rewritten.

Response: Thank you for your comment and we agree that the paragraph lacks clarity. Therefore, we have rewritten it. Specifically, we have rephrased the last sentence of the respective paragraph as “After the index pregnancy, women who got pregnant again were excluded from the present analysis” (L 110-111).

Comment: L 107 what does inclusion refers to here? Use a term commonly used.

Response: Thank you for your comment. We used inclusion to refer to recruitment. As we do not see a difference between the two terms (inclusion and recruitment), we keep ‘inclusion’ in the revised text.

Comment: L 118-132 the writing is not clear at all, correct the grammar please

Response: We have revised the text for grammatical errors.

Comment: L 152 insert the citation for ‘anemia’ and clarify which ‘iron deficiency’ was monitored.

Response: We have cited the reference and rephrased it as iron deficiency anaemia (L 175).

Comment: L 190 insert the source of epi data v 3.1.

Response: Thank you for your comment. We inserted the source (L 211-212).

Comment: L 217-225 were confounding factors considered when analysing the data?

Response: We explored confounding, but we did not see any difference in the results. This is explained it both in the statistical analysis part of the methods section (L 239-247) and in the results section (L 318-324).

Results

Comment: Figure 1 place the y-axis title

Response: Thank you for comment. The title of the Y-axis is missed from the PDF version of the manuscript. We have corrected it as suggested in the revised version.

Comment: Figure 2 caption must be rewritten to make it clearer

Response: We revised the captions of figure 2 as suggested (L 281-285).

Comment: Table 1. Pre-pregnancy BMI was classified into underweight and normal weight. Explain and justify the reason not to tabulate the data based on this sub-cohort.

Response: Tabulating the baseline characteristics by separating the cohorts into underweight and normal weight participants could be an option. Also, tabulating into inadequate and adequate gestational weight gain could be another option. However, as we have high loss to follow up, we prefer to tabulate the study participants in a way that can show the difference in the baseline characteristics between women who successfully completed the study and those dropped out.

Comment: Some parameters are better expressed in range rather than mean values.

Response: Thank you for your suggestion. As most of the variables were normally distributed, we reported mean with standard deviations. Some variables like food insecurity score were not normally distributed. Therefore, we have reported median with interquartile range.

Table 2

Comment: Please clarify whether the baseline characteristics by separating the cohorts into participants completed the study and those dropped out to avoid any skewed data.

Response: Table 2 shows the characteristics of all women in the KITE cohort (column 1), the women who were lost to follow-up between inclusion and measurements at 18 to 24 months postpartum (column 2) and women who were included in the present manuscript (column 3, n=585). We clarified the headers for the columns to better reflect this.

Comment: The footnote must be rewritten to improve understanding, what do participating non-participating women correspond to?

Response: Thank you for your comment. By participating women we meant women who successfully completed the study and by non-participating we meant those dropped out/lost to follow-up. Now we have revised as successfully completed the study vs. lost to follow-up (L 264-266).

Comment: Many qualitative data were collected in the study but only listed out in the tables. The characteristics of the study participants must be included as con-founding factors; this will lead to more meaningful data interpretation.

Discussion

Comment: L 294 how many? N=?

Response: Thank you for your comment. 213 (36.4%) women were undernourished as assessed by BMI. The number is mentioned in the revised text (L 331-335).

Comment: L 296 which studies? Insert citations and were the study cohorts comparable?

Response: We have cited the studies. We do not think that the studies were comparable but there is no similar study in low-income countries. We have shown this in the revised version (L 331-335).

Comment: L 297-297 the statement must include other data, e.g diet diversity, anthropometric and other characteristics before a suggestion was made.

Response: Thank you for your comments. We have revised the text accordingly (L 332).

Comment: L 300-324 as previous comment, data interpretation must take into consideration of other characteristics that could play a role in the weight change.

Response: Several factors were associated with pre-pregnancy nutritional status (Misgina, K.H, et al. BMJ Open 2021, 11, e043484.), and gestational weight gain (Misgina, K.H, et al. BMJ BMC Pregnancy Childbirth 2021, 21, 718). We have discussed some of the evidence presented in these studies in the data interpretation of the present study (L 348-351).

Comment: L 346 insert citation

Response: We have cited the correct reference as suggested (L 386).

Comment: L 355-357 please check for accuracy of the sentence

Response: We think that the sentence is correct. But, we have rephrased it a bit (L 395-397).

Comment: L 361 the baseline data can analyse the confounding factor and see whether they contribute to any difference in the data set.

Response: Thank you for your comment. The contribution of the baseline characteristics is already analysed and reported. However, some of the characteristics might have changed after pregnancy, including  their contribution to weight change. As we do not have specific data concerning these characteristics after pregnancy, we discuss it in the limitation section (L 400-404).

Comment: Authors are advised to clearly describe what the limitations of the study were and suggest improvements in future study.

Response: Thank you for your advice. We have expanded the limitations section a little bit. Also, recommendations for future study are included.

Comment: The conclusion must focus on comprehensively analysed data as per previous comments.

Response: The conclusion is based on our analysis of the association between gestational weight gain and postpartum weight. Higher gestational weight gain (GWG) was associated with higher postpartum weight. Also, a unit increase in GWG was associated with a shift from pre-pregnancy undernutrition (underweight) to normal nutritional status (normal weight) after pregnancy. Furthermore, higher distress was associated with lower likelihood of shifting to normal weight compared to pre-pregnancy nutritional status. Therefore, we think that the conclusion on maternal nutritional status is supported with sufficient results. Given that the inter-pregnancy interval in the study area is short , there is a realistic possibility that women get pregnant again while they still have increased or decreased body weight after their previous pregnancy. Hence, we believe our findings may also be important for future pregnancies.

Reviewer 2 Report

The paper presents an overall interesting look at postpartum weight gain. 

There are several areas that could improve the publication overall: 

Methods: 

The measurement of nutritional status is really difficult to understand. This area must be clarified in subsequent versions. 

Results: 

There are far too many tables. Authors are encouraged to prioritize what is shown in a table, and limit the tables to 2-3, and summarize key features of other tables in the text (and subsequently move the tables to supplemental materials). 

Conclusions:

- The conclusions are overstated; how is the conclusion about the nutritional status determined?

Author Response

Reviewer 2: Comments and suggestions for authors

Comment: The paper presents an overall interesting look at postpartum weight gain. There are several areas that could improve the publication overall:

Response: Thank for your encouraging comment. We have revised the manuscript according to the reviewers comments, particularly the methods section, to make it clearer for the readers.

Methods

Comment: The measurement of nutritional status is really difficult to understand. This area must be clarified in subsequent versions.

Response: Thank you for your comment. The measurement of nutritional status is clarified in detail now in the revised version (L 118-127).

Results

Comment: There are far too many tables. Authors are encouraged to prioritize what is shown in a table, and limit the tables to 2-3, and summarize key features of other tables in the text (and subsequently move the tables to supplemental materials).

Response: Thank you for your suggestion. We have moved table 1 into supplementary materials.

Conclusions

Comment: The conclusions are overstated; how is the conclusion about the nutritional status determined?

Response: (see also the response to the final comment of reviewer 1) Gestational weight gain (GWG) was associated with higher postpartum weight. Also, a unit increase in GWG was associated with a shift from pre-pregnancy undernutrition (underweight) to normal nutritional status (normal weight) after pregnancy. Therefore, we think that the conclusion about nutritional status is supported by sufficient results.